# Effects of SNPs in *AANAT* and *ASMT* Genes on Milk and Peripheral Blood Melatonin Concentrations in Holstein Cows (*Bos taurus*)

**DOI:** 10.3390/genes13071196

**Published:** 2022-07-04

**Authors:** Songyang Yao, Yunjie Liu, Xuening Liu, Guoshi Liu

**Affiliations:** College of Animal Science and Technology, China Agricultural University, Beijing 100193, China; songyangyao@cau.edu.cn (S.Y.); sy20193040665@cau.edu.cn (Y.L.); sy20203040701@cau.edu.cn (X.L.)

**Keywords:** Holstein cows, *AANAT*, *ASMT*, single nucleotide polymorphism (SNP), melatonin milk

## Abstract

Aralkylamine N-acetyltransferase (AANAT) and acetylserotonin O-methyltransferase (ASMT), the two rate-limiting enzymes for melatonin synthesis, regulate melatonin production in mammals. Through analysis of the milk melatonin level and dairy herd improvement (DHI) index, it was found that the melatonin concentration in milk was significantly negatively correlated with the 305 day milk yield (305M) and peak milk yield (PeakM) (*p* < 0.05), while it was significantly positively correlated with the serum melatonin concentration (*p* < 0.05). The full-length of *AANAT* and *ASMT* were sequenced and genotyped in 122 cows. Three SNPs in *AANAT* and four SNPs in *ASMT* were significantly related to MT levels in the milk and serum (*p* < 0.05). The SNPs in *AANAT* were temporarily denoted as N-SNP1 (g.55290169 T>C), N-SNP2 (g.55289357 T>C), and N-SNP3 (g.55289409 C>T). The SNPs in *ASMT* were temporarily denoted as M-SNP1 (g.158407305 G>A), M-SNP2 (g.158407477 A>G), M-SNP3 (g.158407874 G>A), and M-SNP4 (g.158415342 T>C). The M-SNP1, M-SNP2, and M-SNP3 conformed to the Hardy−Weinberg equilibrium (*p* > 0.05), while other SNPs deviated from the Hardy−Weinberg equilibrium (*p* < 0.05). The potential association of MT production and each SNP was statistically analyzed using the method of linkage disequilibrium (LD). The results showed that N-SNP2 and N-SNP3 had some degree of LD (D′ = 0.27), but M-SNP1 and M-SNP2 had a strong LD (D′ = 0.98). Thus, the DHI index could serve as a prediction of the milk MT level. The SNPs in *AANAT* and *ASMT* could be used as potential molecular markers for screening cows to produce high melatonin milk.

## 1. Introduction

Melatonin (MT) is the derivative of tryptophan and was first discovered in 1958 by Lerner et al. from the bovine pineal gland [1]. MT is present in almost every organism, from bacteria to humans, which is directly controlled by light with its high levels during the night [2]. MT is not only produced in the pineal gland, but also in other organs and tissues including the gastrointestinal tract, brain, liver, kidneys, adrenal glands, heart, thymus, gonads, placenta, and uterus [3]. In mammals, its synthesis is from the tryptophan, which is hydroxylated and decarboxylated to form serotonin. Thereafter, the serotonin is subsequently converted to melatonin through the actions of aralkylamine N-acetyltransferase (AANAT) and acetylserotonin O-methyltransferase (ASMT) [4]. AANAT and ASMT are considered to be rate-limiting enzymes in the melatonin synthesis pathway, and both are encoded by *AANAT* and *ASMT* genes, respectively [5,6].

MT has many beneficial effects on human health, including a therapeutic effect on sleep disorders [7,8,9,10]. Thus, studies have focused on improvement of the natural melatonin level in milk. It was found that without artificial light in barns at night, the cows with lower somatic cell counts produced more melatonin-rich milk, and through the use of this diurnal interval, the cows generated functional milk rich in natural melatonin [11]. Melatonin-enriched night milk could increase daytime activity in older adults, which meant they had better rest at night [12]. In addition to the influence of light exposure on melatonin production in milk, we speculate that genetic variation may also impact the melatonin level in milk, especially the single nucleotide polymorphism (SNP).

SNP is just a single base change in a DNA sequence, with a usual alternative of two possible nucleotides at a given position [13]. SNP is considered the most desirable DNA marker for genetic and molecular breeding due to its traceability, ease of interpretation, and applicability to various genotyping techniques [14]. Many studies have shown that livestock wool, milk, meat, and other production performances are strongly associated with SNPs in key regulatory genes [15,16,17,18]. As mentioned above, melatonin-enriched milk was obtained mostly by collecting night milk and controlling the light at night [11,12,19,20], and it ignored the innate melatonin synthesis potential regulated by genetic variations. Using SNP to identify cows with high milk melatonin levels is a novel approach to generate functional milk and it would be more reliable and efficient than that of light regulation.

Thus, in this study, the genetic diversity of the melatonin synthesis rate-limiting enzyme genes of *AANAT* and *ASMT* in Holstein cows was investigated. We expected to find SNP loci that could be significantly associated with bovine melatonin traits, which could be used as molecular markers for screening the natural high melatonin produced by cows.

## 2. Materials and Methods

All experiments were approved by the China Agricultural University Laboratory Animal Welfare and Animal Experimental Ethical Inspection Committee (AW92402202-1-1).

### 2.1. Sample Collection and Genomic DNA Extraction

In this study, 593 milk samples were collected from Farm 1 (Yanqing District, Beijing, China); 122 blood samples and 102 milk samples (some cows were in the dried milk period) were collected from Farm 2 (Changping District, Beijing, China). Milk samples were divided into two portions. One copy was used to detect dairy herd improvement (DHI), which was collected in 50 mL centrifugal tubes with a preservative at 4 °C. Another copy was used to detect MT, which was collected in 15 mL centrifugal tubes without preservative at −20 °C. Blood samples were obtained through the tail vein of the cow and were stored at −20 °C.

Liquid chromatography tandem mass spectrometry (LC-MS/MS) (Santa Clara, CA, USA) was used to detect MT levels in the milk and serum. The C18 column was used to separate the melatonin in the samples. Fossomatic^TM^ FC (serial no. 91755377, part no. 60002326, made in Copenhagen, Denmark) and MilkoScan FT+ (serial no. 91755049, part no. 60027086, made in Copenhagen, Denmark) were used to detect dairy herd improvement (DHI) which included days in milk yield (DIM), lactation (L#), herd test milk yield (HTM), herd test adjusted corrected milk yield (HTACM), previous milk yield (Prev.M), fat percentage (FP), protein percentage (PP), fat/protein (F/P), lactose percentage (LP), solid percentage (SP), somatic cell count (SCC), milk loss (Mlos), logarithm with somatic cell count (LSCC), previous SCC (PreSCC), first grade SCC (1stSCC), lactation to date milk (LTDM), 305 days milk yield (305M), peak milk yield (PeakM), peak day (PeakD), and persist and lactation length (LacLen).

The TsingKe Genomic DNA Extraction Kit (TSP201-200, Beijing, China) was used to extract the genomic DNA. Extraction quality was judged using a Nano-300 spectrophotometer (China) on the 1.8 < OD260/280 < 2.0 standard. All of the qualified genomic DNA was diluted and placed at −20 °C for storage.

### 2.2. Primer Design

Primers were designed from the NCBI database for the bovine (*Bos taurus*) *AANAT* gene sequence (GenBank accession No. NC_037346.1) and the *ASMT* gene sequence (GenBank accession no. NC_037328.1). Primer Premier 6.0 software (Quebec City, QC, Canada) was used to design the primers for both genes to amplify the full length of both genes. Three primers were designed for *AANAT*, and thirteen primers were designed for *ASMT.*

### 2.3. SNP Screening and SNP Genotyping

The PCR products were verified by 1.0% agarose gel electrophoresis, and those that matched the single zone were sent to the Beijing TsingKe Biotechnology Company for Sanger sequencing. The sequence results were viewed comparatively using Chromas software (Version 2.5), and SNPs were genotyped using TSINGKE SNP1 software to screen SNPs. SNPs were identified by the presence of a double peak at each unit locus in the chromatogram. After that, quality control analysis was performed on the candidate SNPs, and the qualified SNPs were screened.

PCR amplification was performed in the 25 μL reaction system, including 1 μL genomic DNA (20 ng/μL), 1 μL of forward/reverse primers (10 μM), and 22 μL of 1.1× T3 Super PCR Mix (TsingKe). The PCR amplification protocol contained a pre-denaturation stage at 95 °C for 5 min, and 35 cycles of circulation stage were 98 °C for 10 s, 58–61 °C for 10 s, and 72 °C for 10 s. A final extension stage was 72 °C for 5 min with subsequent cooling to 4 °C.

## 3. Statistical Analysis

The genotype and allele frequencies, homozygosity (Ho), and expected heterozygosity (He) were calculated by Excel software (Version 2021). The allele frequencies of each SNP were calculated for departure from the Hardy−Weinberg equilibrium (HWE) by using the chi-square test (χ^2^-test). SNP−phenotype correlations were analyzed using R language software. According to the source of the cow sample, for lactation, 305M, and PeakM, a statistical analysis model was established as
Y*_ijkl_* = μ + G*_i_* + L*_j_* + M*_k_* + P*_l_* + e*_ijkl_*
(1)
in which Y*_ijkl_* refers to the measured value of the phenotypic trait, μ represents the population mean, G*_i_* is the genotype effect, L*_j_* is the lactation effect, M*_k_* is the 305M effect, P*_l_* is the PeakM effect, and e*_ijkl_* is the random residual. The sequencing samples were collected from the same cattle farm partition and the same time, so no farm effects and time effects were added to our linear model. A correlation analysis between phenotypes was performed using SPSS 22.0 software. The phenotypes under different genotypes of each SNP locus were analyzed using analysis of variance (ANOVA) and multiple comparisons. PLINK software was used to analyze the linkage disequilibrium (LD) between SNPs.

## 4. Results

### 4.1. Distribution of MT Levels in Milk of 695 Cows 

First, 695 milk samples were collected from Farm 1 (*n* = 593) and Farm 2 (*n* = 102) at the same period. The results showed that the MT levels in milk were wildly distributed and ranged from 1.43 pg/mL to 7.86 ng/mL among the tested cows. As shown in Figure 1, 443 milk samples were less than 50 pg/mL, accounting for 63.74%; 132 milk samples were 50–100 pg/mL, accounting for 18.99%; 89 milk samples were 100–500 pg/mL, accounting for 12.8%; and 31 milk samples were more than 500 pg/mL, accounting for 4.46%.

### 4.2. Correlation Analysis between Milk MT Levels and the Index of DHI

The potential association between MT level and DHI in 593 milk samples from Farm 1 were analyze. The results are showed in Table 1, which indicates that the MT level in the milk was significantly negatively correlated with 305M (*p* = 0.041) and PeakM (*p* = 0.011) and also negatively correlated with L# (*p* = 0.050). The Pearson correlation coefficients r were −0.092, −0.104, and −0.081, respectively. 

### 4.3. Association between Milk and Serum MT Levels

The result of the 102 serum and milk samples showed that MT in milk was significantly positively correlated with MT in serum (*p* = 0.036). The Pearson correlation coefficient r was 0.21.

### 4.4. Identification of Three Significantly Associated SNPs within AANAT and Four SNPs within ASMT in Cows

We identified 25 SNPs in the coding sequence (CDS) region of *AANAT*, of which 14 SNPs caused amino acid mutations. There were nine missense mutations and five synonymous mutations among the 14 SNPs. Two SNPs in the coding sequence (CDS) region of *ASMT* were identified, both of which were amino acid synonymous mutations. Neither of these two gene CDS region SNPs were significantly correlated with the MT level. Therefore, we proceeded to analyze the correlation of the non-coding region SNPs of these two genes with the melatonin levels. The correlation analysis revealed that three SNPs within *AANAT* and four SNPs within *ASMT* were significantly associated with MT in the milk and serum based on the DNA sequencing and sequence alignments results from 122 cows. SNPs that were significantly associated with MT levels in milk within *AANAT* were denoted as N-SNP1 (g.55290169 T>C) (*p* = 0.0155), and those significantly associated with serum melatonin were denoted as N-SNP2 (g.55289357 T>C) (*p* = 0.042) and N-SNP3 (g.55289409 C>T) (*p* = 0.0142). SNPs that were significantly associated with MT in milk within *ASMT* were denoted as M-SNP1 (g.158407305 G>A) (*p* = 0.0332), M-SNP2 (g.158407477 A>G) (*p* = 0.0363), and M-SNP3 (g.158407874 G>A) (*p* = 0.0332), and being significantly associated with MT in the serum was denoted as M-SNP4 (g.158415342 T>C) (*p* = 0.046). N-SNP1 and N-SNP3 are located in the *AANAT* exon region and N-SNP2 is located in the intron region. M-SNP1, M-SNP2, and M-SNP3 are located in the ASMT intron region and M-SNP4 is located in the exon region. The sequence chromatograms of the heterozygous genotypes are illustrated in Figure 2.

### 4.5. Genotypic and Allelic Frequencies and Genetic Parameter

Table 2 shows the genotype frequencies, allelic gene frequencies, Ho, He, and HWE of seven SNPs within the *AANAT* and *ASMT* genes in cows. After the chi-square test, N-SNP1, N-SNP2, N-SNP3, and M-SNP4 deviated from the Hardy−Weinberg equilibrium. No homozygous mutation genotype was detected in M-SNP4. The wild allele frequencies of all SNPs were more than those of the mutation alleles.

### 4.6. MT Levels in the Serum and Milk under Different Genotypes of SNP

The milk MT levels of the homozygous mutation genotype CC (23.67 ± 6.64 pg/mL) in N-SNP1 were significantly lower compared with the TT (44.72 ± 2.99 pg/mL) and CT (49.93 ± 5.56 pg/mL) genotypes within *AANAT* (*p* < 0.05) (Figure 3A). The milk MT levels of the heterozygote mutation genotype AG (51.69 ± 3.80 pg/mL) in M-SNP1 were significantly higher compared with the AA (37.13 ± 5.79 pg/mL) genotype within *ASMT* (*p* < 0.05) (Figure 3B). The milk MT levels of the heterozygote mutation genotype AG (51.31 ± 3.90 pg/mL) in M-SNP2 were significantly higher compared with the GG (38.97 ± 5.70 pg/mL) genotype within *ASMT* (*p* < 0.05) (Figure 3C). No significant difference was found in the milk MT levels in M-SNP3 within *ASMT* (Figure 3D).

The serum MT levels were significantly higher in the homozygous mutation genotype CC (166.59 ± 32.63 pg/mL) in N-SNP2 compared with the TT (102.24 ± 6.53 pg/mL) and CT (99.94 ± 5.82 pg/mL) genotypes within *AANAT* (*p* < 0.05) (Figure 4A). No significant difference was found in the serum MT levels in N-SNP3 within *AANAT* (Figure 4B). The MT levels were significantly lower in the serum of the heterozygote mutation genotype CT (107.13 ± 8.84 pg/mL) in M-SNP4 compared with the CC (155.73 ± 29.58 pg/mL) genotypes within *ASMT* (*p* < 0.05) (Figure 4C).

### 4.7. Linkage Disequilibrium Analysis of SNP Loci within Cow AANAT and ASMT Genes

The three SNPs within *AANAT* and four SNPs within *ASMT* were significantly associated with the MT levels in cows. Is there any correlation among these SNP loci within the same gene? To further investigate this issue, linkage disequilibrium (LD) analysis was performed on different SNPs within the same gene (Table 3). N-SNP2 had a closer LD relationship with N-SNP3 than the other SNPs loci within *AANAT* (D′ = 0.27). M-SNP1 had a stronger LD relationship with M-SNP2 than the other SNP loci within *ASMT* (D′ = 0.98). These close links among four loci within *AANAT* and *ASMT* suggested that there might be a certain synergistic effect on regulating the MT production traits of cows. 

## 5. Discussion

Because of its increased demand, melatonin-enriched milk has become popular in the international market in recent years. Milk MT is a natural source of MT and exhibits many beneficial effects on human health [10,11]. Therefore, the price of MT-enriched milk is almost three-fold higher than that of regular milk [21]. Studies have shown that individuals are willing to pay more to buy night milk (melatonin-enriched) to improve their sleep quality [22,23].

The current MT-enriched milk is mostly produced by either controlling light exposure period during the night or the collection of milk at night [12,24]. Some studies have also shown that feeding rumen-protected tryptophan or feeding a high melatonin content of mulberry leaves and soybeans also increases milk MT [25,26]. In our study, we found that at the same feeding condition, 82.73% of cows had milk MT concentrations below 100 pg/mL, and only 4.46% of cows had milk MT in excess of 500 pg/mL. This indicated that some naturally high MT producing cows do exist, independent of environmental conditions (Figure 1). After correlating the raw milk MT concentrations with the serum MT and DHI indicators, milk MT was positively correlated with serum melatonin and negatively correlated with 305M and PeakM. As we know, in mammals, their melatonin levels decline with aging [27]. Therefore, we initially selected possible natural high MT populations from cows that had not given birth much, had low 305M, and had PeakM phenotypes. The 305M, PeakM, and number of births were added as fixed effects in the analytical model to simulate the potential correlation between the SNP and MT levels. As a result, we believe that these correlations are reliable and without the influence of the aging. We also realized that the overexpression of the *AANAT* and *ASMT* in dairy goats increased the melatonin levels in peripheral blood and milk, while improving milk quality [28,29,30].

As the results indicated in this study, the homozygous mutation genotype of N-SNP1 located in the exon had significantly lower milk MT than other genotypes, and the homozygous mutation genotype of N-SNP2 located in the intron had significantly higher serum MT than the other genotypes. While the heterozygote mutation genotype of M-SNP1 and M-SNP2 located in the intron had significantly higher milk MT than the other genotypes, and the heterozygote mutation genotype of M-SNP4 in the intron had significantly lower serum MT than the other genotypes. The mutations in the exons may directly impact the activities of the enzymes, while the mutations in the intron may regulate the expressions of *AANAT* and *ASMT*. It seems that some SNPs have an association with serum melatonin level and others have an association with the milk melatonin level. Actually, the serum and milk melatonin levels are positively correlated. This was supported by the findings that the different SNPs were associated with each other related to melatonin production, and other studies also confirmed that SNPs affect the economic traits of animals, such as the litter size and growth performance of goats [31,32]. For example, M-SNP1 and M-SNP2 had a strong LD relationship (D′ = 0.98), and the D′ value (0.27) between N-SNP2 and N-SNP3 was close to the threshold of the strong linkage disequilibrium state (D′ = 0.33), suggesting these two SNPs within *AANAT* and *ASMT* may function synergistically in influencing serum and milk melatonin levels in cows. In the current study, we could not justify the exact mechanisms of how the SNPs within *AANAT* and *ASMT* modify the milk and serum melatonin levels, but the influence of these SNAP on melatonin production does exist. Clarifying their molecular mechanisms is a goal for future studies.

These SNPs of *AANAT* and *ASMT* identified in the current study could serve as potential molecular markers for screening cows who have high levels of natural melatonin in milk. With this new approach, the production of natural melatonin-enriched milk will not be limited to the collection of night milk and feed conditions. The genetic breeding and selection will establish the core herd of natural high melatonin cows. The SNPs on the melatonin synthesis rate-limiting enzyme genes *AANAT* and *ASMT* obtained in this study laid the foundation for the selection of the core herd of naturally high melatonin cows.

## Figures and Tables

**Figure 1 genes-13-01196-f001:**
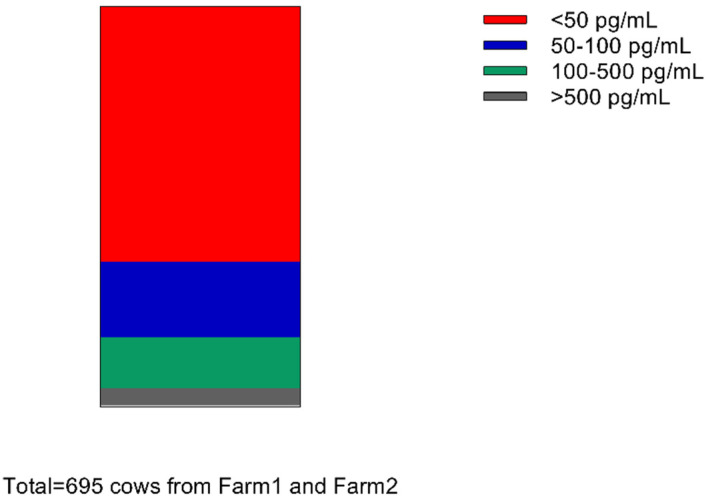
MT level distribution of 695 milk samples.

**Figure 2 genes-13-01196-f002:**
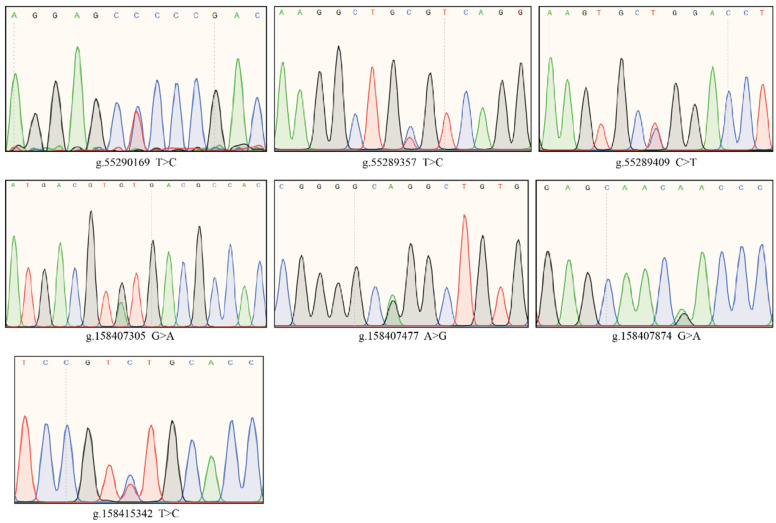
The sequence chromatograms of the heterozygous genotypes.

**Figure 3 genes-13-01196-f003:**
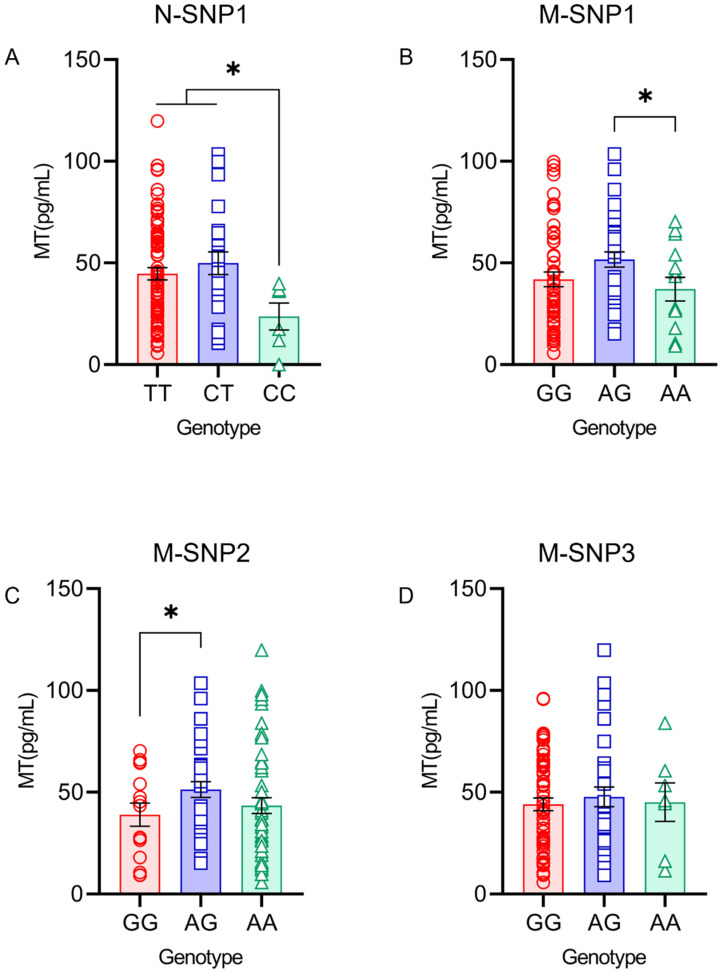
MT levels in milk under different genotypes within the *AANAT* and *ASMT* genes. * (*p* < 0.05) show significant difference. (**A**) Milk MT level of N-SNP1 within *AANAT*. (**B**) Milk MT level of M-SNP1 within *ASMT*. (**C**) Milk MT level of M-SNP2 within *ASMT*. (**D**) Milk MT level of M-SNP3 within *ASMT*.

**Figure 4 genes-13-01196-f004:**
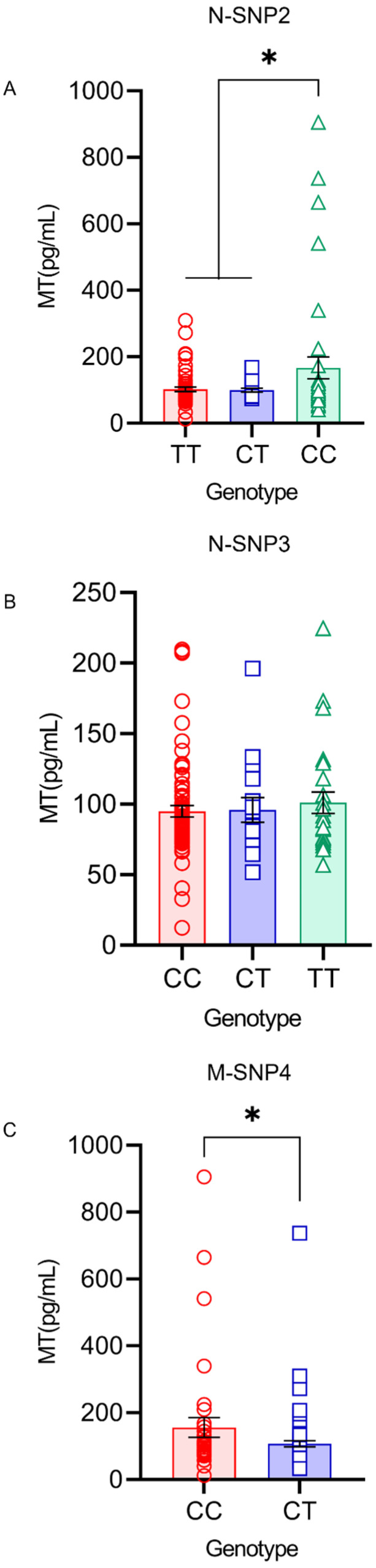
MT levels in the serum under different genotypes within the *AANAT* and *ASMT* genes. * (*p* < 0.05) show a significant difference. (**A**) Serum MT level of N-SNP2 within *AANAT*. (**B**) Serum MT level of N-SNP3 within *AANAT*. (**C**) Serum MT level of M-SNP4 within *AANAT*.

**Table 1 genes-13-01196-t001:** Pearson correlation analysis between MT in milk and DHI index.

DHI Index	Sample Size	Pearson r	*p* Values
DIM	593	−0.013	0.754
L#	593	−0.081	0.050
HTM	593	−0.055	0.185
HTACM	593	−0.032	0.438
Prev.M	593	−0.070	0.105
FP	540	−0.007	0.873
PP	593	−0.017	0.680
F/P	593	−0.011	0.781
LP	593	0.008	0.855
SP	593	−0.009	0.824
SCC	593	0.017	0.679
MLos	593	−0.025	0.539
LSCC	593	−0.037	0.374
PreSCC	593	−0.035	0.418
1stSCC	539	−0.031	0.455
LTDM	593	−0.037	0.364
305M	593	−0.092	0.041 *
PeakM	492	−0.104	0.011 *
PeakD	593	0.002	0.961
Persist	593	0.015	0.728
LacLen	540	−0.013	0.754

* show significant difference, *p* < 0.05.

**Table 2 genes-13-01196-t002:** The genetic diversity of the cow *AANAT* and *ASMT* genes.

SNP Loci	Genotypic Frequencies			Allele Frequencies		Genetic Parameter		
	D	H	R	Reference	Mutation	Ho	He	HWE (*p* Value)
N-SNP1	0.706 (72)	0.225 (23)	0.069 (7)	0.819	0.181	0.704	0.296	0.0484 *
N-SNP2	0.508 (62)	0.18 (22)	0.312 (38)	0.598	0.402	0.519	0.481	<0.00001 *****
N-SNP3	0.623 (76)	0.148 (18)	0.229 (28)	0.697	0.303	0.578	0.422	<0.00001 *****
M-SNP1	0.51 (52)	0.343 (35)	0.147 (15)	0.681	0.319	0.566	0.434	0.1079
M-SNP2	0.51 (52)	0.333 (34)	0.157 (16)	0.676	0.324	0.562	0.438	0.0547
M-SNP3	0.569 (58)	0.353 (36)	0.078 (8)	0.745	0.255	0.62	0.38	0.7658
M-SNP4	0.303 (37)	0.697 (85)	NA	0.65	0.35	0.545	0.455	0.000051 ****

Genotype frequency: D, homozygous wild type genotype; H, heterozygous mutant genotype; R, homozygous mutant genotype; Ho, homozygosity; He, heterozygosity; HWE, Hardy–Weinberg equilibrium. * shows a significant difference, *p* < 0.05. **** shows an extremely significant difference, *p* < 0.0001. ***** shows an extremely significant difference, *p* < 0.00001.

**Table 3 genes-13-01196-t003:** Linkage disequilibrium (LD) analysis of seven SNP loci within the cow *AANAT* and *ASMT* genes.

Gene	*AANAT*			*ASMT*					
LD	N-SNP1 N-SNP2	N-SNP1 N-SNP3	N-SNP2 N-SNP3	M-SNP1 M-SNP2	M-SNP1 M-SNP3	M-SNP1 M-SNP4	M-SNP2 M-SNP3	M-SNP2 M-SNP4	M-SNP3 M-SNP4
D′	0.004	0.009	0.27	0.98	0.05	0.0002	0.05	0.0005	0.0007

## Data Availability

Not applicable.

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
