# Peer review of "Effects of SNPs in AANAT and ASMT Genes on Milk and Peripheral Blood Melatonin Concentrations in Holstein Cows (Bos taurus)"

_genes, 2022, doi:10.3390/genes13071196_

Round 1

Reviewer 1 Report

Line 30: "wa5s" to was

Line 32: "almost" has to be removed

Line 36: change "was" to "is"

Line 41: "to improve" should be "the improvement"

Line 49 should be rephrased

Lines 60-62 should be rephrased

Line 80 should be rephrased

Line 88 should be rephrased

Why did you pull the DNAs for variant detection?

How did you compute Genotype and allele frequencies, homozygosity (Ho), and expected heterozygosity (He) from the pooled samples?

The results are not clear. You should provide more details for the association analyses

Author Response

  1. Line 30: "wa5s" to was

Changed

  1. Line 32: "almost" has to be removed

Changed

  1. Line 36: change "was" to "is"

Changed

  1. Line 41: "to improve" should be "the improvement"

Changed

  1. Line 49 should be rephrased

Changed to: SNP is just a single base change in a DNA sequence, with a usual alternative of two possible nucleotides at a given position.

  1. Lines 60-62 should be rephrased

Changed to: We expected to find SNP loci that could be significantly associated with bovine melatonin traits.

  1. Line 80 should be rephrased

Changed to: TsingKe Genomic DNA Extraction Kit (TSP201-200, China) was used to extract genomic DNA.

  1. Line 88 should be rephrased

Changed to: Primer Premier 6.0 software was used to design primers for both genes to amplify the full length of both genes. Three primers were designed for AANAT, and thirteen primers were designed for ASMT.

  1. Why did you pull the DNAs for variant detection?

I have a problem with my expression. What I mean is to detect SNPs by comparing them with reference sequences.

  1. How did you compute Genotype and allele frequencies, homozygosity (Ho), and expected heterozygosity (He) from the pooled samples?

The Genotypic frequencies were computed by the following formula.

D, homozygous wild type genotype=Number of homozygous wild genotypes/The total number of samples

H, heterozygous mutant genotype=Number of heterozygous mutant genotypes/The total number of samples

R, homozygous mutant genotype=Number of homozygous mutant genotypes/ The total number of samples

The allele frequencies were computed by the following formula.

Reference=(2* number of homozygous wild genotypes + Number of heterozygous mutant genotypes)/2* The total number of samples

Mutation=(2* number of heterozygous mutant genotypes + Number of heterozygous mutant genotypes)/2* The total number of samples

The Ho and He were computed based on Hardy Weinberg equilibrium.

A=Reference, B=Mutation

Ho=A2+B2, He=2AB

  1. The results are not clear. You should provide more details for the association analyses

I have changed the Figure between DHI and MT level in milk to Table with N, correlation coefficient r and p value.

The error in the description of the result has been corrected.

Some results has been rephrased.

Reviewer 2 Report

The paper deals with the SNPs in the AANAT and ASMT genes and melatonin in cows. The topic is of interest and the manuscript brings new knowledge. However, there are some imperfections, which must be revised.

MM section

This part is rather scanty, please describe the methods better.

You write on milk samples, how was the count of cows? Or is it the same?

There are abbreviations without explanation, DHI, L#, etc. What is herd test milk, herd test adjusted corrected, Mlos, etc.

How have you chosen the cows into the pools (rr. 91-92).

Have you tested other effects (r. 113), which effects were significant?

At which time of day have you sampled the milk and blood for melatonin analysis?

Results

You have found just three and four SNPs with significant correlation to melatonin (rr.40-142). Please, give some information on the other SNPs which you have found.

I do not like the Fig. 2. I would prefer the table with n, correlation coefficients and p values.

The same for Fig. 3. There is not the reference on the figure in the text.

You write on six SNPs (r. 155), but there are seven SNPs in the Table 1?

From HWE deviated N-SNP1, N-SNP2, N-SNP3 and M-SNP4? Please, sign the significance by asterisk * in the Tab. 1.

Reformulate the sentence “Six…and ASMT.” (rr. 158-159), it is a bit tortuous.

Consider adding of the means up the columns in the Figs 5,6.

Discussion

Reformulate the sentence “Studies…23].” (rr. 186-187).

Formal errors:

1st word in Abstract in bold.

Check the font size in Abstract.

R. 30 “was”.

R. 42 twice “that”.

In Figs 5,6, genotype, not genetype.

Author Response

The paper deals with the SNPs in the AANAT and ASMT genes and melatonin in cows. The topic is of interest and the manuscript brings new knowledge. However, there are some imperfections, which must be revised.

MM section

  1. This part is rather scanty, please describe the methods better.

Added details of milk sampling and details of blood samples.

  1. You write on milk samples, how was the count of cows? Or is it the same?

We collected 593 milk samples from 593 cows from Farm1, and we collected 102 milk samples from 122 cows because some cows were in the dried milk period.

  1. There are abbreviations without explanation, DHI, L#, etc. What is herd test milk, herd test adjusted corrected, Mlos, etc.

I had add the explanation in Line73-80.

DHI: Dairy Herd Improvement

L#: Lactation

Herd Test Milk: Milk yield daily.

Herd test adjusted corrected: Daily milk yield after correction of coefficients.

Mlos: Milk loss

  1. How have you chosen the cows into the pools (rr. 91-92).

I have a problem with my expression. I collected all cows’ samples from Farm 1 and Farm 2.

  1. Have you tested other effects (r. 113), which effects were significant?

I considered the reasons of time and season. However, I collected them in the same season and time period. Therefore, I did not put them into the analysis model. I also explained this reason on Line113-115.

  1. At which time of day have you sampled the milk and blood for melatonin analysis?

I collected the milk and blood on 6:00-9:00 AM of day.

Results

  1. You have found just three and four SNPs with significant correlation to melatonin (rr.140-142). Please, give some information on the other SNPs which you have found.

The other information of SNPs was added in Result 4.4.

Added information: We identified 25 SNPs in the Coding sequence (CDS) region of AANAT, of which 14 SNPs caused amino acid mutations. 9 missense mutations and 5 synonymous mutations among the 14 SNPs. Two SNPs in the Coding sequence (CDS) region of ASMT were identified, both of which were amino acid synonymous mutations. Neither of these two gene CDS region SNPs was significantly correlated with MT level. Therefore, we proceeded to analyze the correlation of Non-coding region SNPs of these two genes with melatonin levels.

  1. I do not like the Fig. 2. I would prefer the table with n, correlation coefficients and p values.

I have changed the Fig.2 to Table1, which showed the N, correlation coefficients and p values.

  1. The same for Fig. 3. There is not the reference on the figure in the text.

The result 4.3 showed the correlation between milk in serum and milk in serum. I considered simply deleting Figure 3 and describing the results directly.

  1. You write on six SNPs (r. 155), but there are seven SNPs in the Table 1?

Changed

  1. From HWE deviated N-SNP1, N-SNP2, N-SNP3 and M-SNP4? Please, sign the significance by asterisk * in the Tab. 1.

Changed

  1. Reformulate the sentence “Six…and ASMT.” (rr. 158-159), it is a bit tortuous.

Changed to: The wild alleles frequencies of all SNPs were more than that of mutation alleles.

  1. Consider adding of the means up the columns in the Figs 5,6.

I don't really understand the meaning of this comment, but I modified the Figure.

Discussion

  1. Reformulate the sentence “Studies…23].” (rr. 186-187).

Changed to: Studies have showed that individuals were willing to pay more to buy night milk (melatonin-enriched) to improve their sleep quality

Formal errors:

  1. 1st word in Abstract in bold.

Changed

  1. Check the font size in Abstract.

Changed

  1. R. 30 “was”.

Changed

  1. R. 42 twice “that”.

Changed

  1. In Figs 5,6, genotype, not genetype.

Changed